# Identification and Functional Analysis of the Regulatory Elements in the p*HSPA6* Promoter

**DOI:** 10.3390/genes13020189

**Published:** 2022-01-21

**Authors:** Shuyu Jiao, Chunyan Bai, Chunyun Qi, Heyong Wu, Lanxin Hu, Feng Li, Kang Yang, Chuheng Zhao, Hongsheng Ouyang, Daxin Pang, Xiaochun Tang, Zicong Xie

**Affiliations:** 1College of Animal Science, Jilin University, Changchun 130062, China; jiaosy0105@163.com (S.J.); bcy@jlu.edu.cn (C.B.); qicy21@mails.jlu.edu.cn (C.Q.); heyong21@mails.jlu.edu.cn (H.W.); hulx20@mails.jlu.edu.cn (L.H.); lifeng21@mails.jlu.edu.cn (F.L.); kangyang20@mails.jlu.edu.cn (K.Y.); zch20@mails.jlu.edu.cn (C.Z.); ouyh@jlu.edu.cn (H.O.); pdx@jlu.edu.cn (D.P.); 2Key Lab for Zoonoses Research, Ministry of Education, Animal Genome Editing Technology Innovation Center, Jilin University, Changchun 130062, China; 3Chongqing Research Institute, Jilin University, Chongqing 401123, China; 4Chongqing Jitang Biotechnology Research Institute Co., Ltd., Chongqing 401123, China

**Keywords:** HSP70, pig, p*HSPA6*, heat shock element, stress response

## Abstract

Functional and expressional research of heat shock protein A6 (HSPA6) suggests that the gene is of great value for neurodegenerative diseases, biosensors, cancer, etc. Based on the important value of pigs in agriculture and biomedicine and to advance knowledge of this little-studied HSPA member, the stress-sensitive sites in porcine *HSPA6* (p*HSPA6*) were investigated following different stresses. Here, two heat shock elements (HSEs) and a conserved region (CR) were identified in the p*HSPA6* promoter by a CRISPR/Cas9-mediated precise gene editing strategy. Gene expression data showed that sequence disruption of these regions could significantly reduce the expression of p*HSPA6* under heat stress. Stimulation studies indicated that these regions responded not only to heat stress but also to copper sulfate, MG132, and curcumin. Further mechanism studies showed that downregulated p*HSPA6* could significantly affect some important members of the HSP family that are involved in HSP40, HSP70, and HSP90. Overall, our results provide a new approach for investigating gene expression and regulation that may contribute to gene regulatory mechanisms, drug target selection, and breeding stock selection.

## 1. Introduction

The heat shock protein 70 (HSP70) family is a set of highly conserved proteins that are known for increased synthesis, exposing organisms to a mass of cellular stresses. More recently, a vast array of post-translational modifications (PTMs) of HSP70 family proteins has been uncovered, involving altering chaperone activity, localization, and selectivity [1]. *HSPA6* is a poorly characterized member of the HSP70 family and is induced after severe cellular stress [2,3]. Heat stress is not only detrimental to animal production and reproduction but also seriously damages the health of breeding animals [4]. *HSPA6* was identified to have higher expression in cattle and goats under heat stress [5,6,7]. This may be because severe stress conditions cause *HSPA6* to evolve into a gene that maintains basic biological functions [8]. Therefore, *HSPA6* can be used as a candidate gene for breeding heat-resistant herds [7,9]. In addition, a previous study indicated that Parkinson’s disease and Alzheimer’s disease (AD) led to 30.4-fold and 6.3-fold increases, respectively, in *HSPA6* expression [10,11,12]. HSPA6 was strongly induced by MG132 treatment of human nerve cells [13,14]. Knockdown experiments showed that HSPA6 could protect human nerve cells from stress [15]. Therefore, the results implied that human nerve cells treated with MG132 could be used as cell models for studying neurodegenerative diseases. Furthermore, HSPA6 was induced and functioned as a positive regulator for Enterovirus A71 infection, indicating that HSPA6 may assist the function of a cellular protein generally required for viral IRES activities [16]. In fact, recent studies have shown that *HSPA6* is also involved in cancer progression, including that of lung cancer, triple-negative breast cancer (TNBC), and bladder cancer [17,18,19]. Interestingly, the *HSPA6* promoter is also widely used in the biomedical field because of its ability to control cell activity at specific sites. For example, engineered T cells can control gene expression remotely [20]. Moreover, recent findings used the *HSPA6* promoter to construct a cell line that can evaluate cell sensors in response to a stimulus based on EGFP expression [21].

*HSPA6* was cloned as a novel HSP70 gene that is different from others and expressed in many organisms [22]. There is no *HSPA6* homolog in rodents, but *HSPA6* exists in higher mammals, such as humans, goats, and swine [5,23,24]. In addition to expression among limited species, *HSPA6* is barely expressed in many cells under normal culture conditions. The mRNA expression level of *HSPA6* has a large increase under various cellular stresses, such as mechanical [3], copper and zinc [25], nitric oxide [26,27], heavy metal [28], and antibiotic [29] stresses. Additionally, constitutive expression of *HSPA6* has been explained through the proximal promoter [30]. Deletion of the DNA fragment (−346 to −217 bp) resulted in the loss of most promoter basal activation. Furthermore, a previous assay using the DNA fragment from −287 to +110 of the *HSPA6* gene (accession no. AL590385) revealed cadmium chloride-induced cytotoxicity with fourfold higher sensitivity than a cell viability test. Then, tandem repeats of the cytotoxic responding element (CRE) in the *HSPA6* promoter were used to construct highly sensitive cytotoxicity sensor cells [31]. Meanwhile, some transcriptional elements were predicted, and their effect on the expression of *HSPA6* was tested. AP1 plays more roles in the constitutive and inducible expression of *HSPA6*, whereas HSF regulates *HSPA6* transcription under stress conditions [32].

After briefly summarizing the *HSPA6* context, these studies mainly focused on *HSPA6* promoter characterization in humans rather than pigs, which can serve as biomedical models. To further explore the p*HSPA6* promoter, we identified the regulatory elements in the p*HSPA6* promoter and validated their functions based on our previously established p*HSPA6*-based EGFP fluorescent reporter cell line using precise gene editing strategies mediated by CRISPR/Cas9 [21].

## 2. Materials and Methods

### 2.1. Cell Culture and Treatment

Porcine primary kidney cells (PK), which were isolated from 2-day-old fetuses, porcine kidney cell lines (PK-15) (Lot Number: 58808810 ATCC Number: CCL-33), and H_2_E were cultured in Dulbecco’s modified Eagle’s medium (DMEM) supplemented with 5% fetal bovine serum (FBS) (Gibco, Grand Island, NY, USA). Porcine fetal fibroblasts (PFFs) were isolated from 33-day-old fetuses, and porcine alveolar macrophage cell lines (3D4/2) were obtained from Prof. Li Feng of Harbin Veterinary Research Institute, and the cells were grown in DMEM supplemented with 15% FBS (Gibco, Grand Island, NY, USA). All cells were incubated at 37 °C in an atmosphere of 5% CO_2__,_ and the medium was changed every 2 days.

Cells were exposed to 42 °C for 100 min and recovered at 37 °C with 5% CO_2_. Concentrations of 500 μM copper sulfate, 30 μM MG-132, and 40 μM curcumin were used to stimulate cells for 100 min and cultured under standard conditions (DMEM supplemented with 5% FBS) after washing with PBS buffer solution. Finally, cells were collected at 0, 1, 2, 4, 8, 12, and 24 h post-heat/treatment for follow-up experiments.

### 2.2. Plasmids

sgRNAs (listed in Table 1) targeting the three regions of the p*HSPA6* promoter were designed by online software (http://crispr.mit.edu/, accessed on 10 December 2020) and synthesized by Comate Bioscience Co., Ltd. (Changchun, China). Targeting sgRNAs were ligated to the PX330 vector (42230, Addgene) using the method [33] described by Zhang at the Broad Institute of MIT.

Four complementary sgRNA oligo DNAs were synthesized and annealed to double-stranded DNA in the presence of 10 × NEB standard Taq buffer, and the whole targeting plasmids formed using this product were ligated into the BbsI sites of the vector backbone (PX330).

### 2.3. Electroporation of H_2_E Cells

For electroporation, approximately 30 µg of plasmids and 3 × 10^6^ H_2_E cells were suspended in 300 μL of Opti-MEM (Gibco, Grand Island, NY, USA) in 2 mm gap cuvettes and electroporated using specified parameters with a BTX-ECM 2001. The electroporation parameters for H_2_E cells were as follows: 280 volts (V), 1 millisecond (ms), and 3 pulses for 1 repeat.

### 2.4. Selection of Cell Clones

At 48 h posttransfection, H_2_E cells were inoculated into ten 100 mm dishes at an appropriate density (1000–2000 cells/dish). Single-cell clones were picked and inoculated in 24-well plates. When the confluence reached 80% or more, 15% of each clone was digested and lysed with 10 μL NP40 lysis buffer (0.45% NP40 plus 0.6% proteinase K) for 1 h at 56 °C and 10 min at 95 °C. The lysate was used as the PCR template and was subjected to 1% agarose gel electrophoresis. To confirm the genotype of cell clones, pLB vectors (Tiangen, Beijing, China) were combined with some PCR products, and constructors were sequenced to determine the exact sequences. Finally, the cell clones were frozen in liquid nitrogen, and the locations were recorded.

### 2.5. PCR Detection

We performed PCR to test cell clones for site-specific knockout of CR. The primers are listed in Table 2. The PCR conditions were 95 °C for 4 min; 94 °C for 30 s, 58 °C for 30 s, 72 °C for 40 s, for 32 cycles; 72 °C for 5 min; hold at 16 °C; and the polymerase was 2× Taq PCR Mastermix (Tiangen, Beijing, China).

### 2.6. Quantitative Real-Time PCR Analysis

RZ reagent (Tiangen, Beijing, China) was used to extract total RNA. First-strand cDNAs were generated through reverse transcription using total RNA and oligo-dT primers. Porcine *GAPDH* served as the reference gene for the relative expression of p*HSPA6*. The 2−ΔΔCt method was performed for mRNA expression, which was standardized to the expression of endogenous U6. Primers are listed in Table 2.

### 2.7. Fluorescence Microscopy and Flow Cytometric Analysis

These positive clones were treated with stressors at 100 min. After 1 h incubation at 37 °C, cells were washed three times with PBS and then incubated with Hoechest 33342 diluted at 1 to 1000 for 6 min at 37 °C. After three washes with PBS, the cells were examined under a fluorescence microscope (Olympus BX51, Tokyo, Japan). The harvested cells were washed twice, resuspended in 300 µL of DPBS, and analysed using a BD Accuri C6 flow cytometer.

### 2.8. Statistical Analysis

The data were statistically analyzed with GraphPad Prism software (*t* test), and all values were considered statistically significant at *p <* 0.05. 

## 3. Results

### 3.1. Expression of HSPA6 in Porcine Cells

To measure the expression of p*HSPA6*, we studied the effects on p*HSPA6* expression in different types of porcine cells. Constitutive and inducible expression of p*HSPA6* was examined in PK-15, PFF, and 3D4/2 cells at 37 °C and 42 °C. Constitutive expression of p*HSPA6* was detected with 2977.42-, 20,190-, and 48.87-fold increases in mRNA levels post heat stress, respectively (Figure 1a). At the same time, we analysed fluorescence intensity in EGFP-KI reporter PK-15 and PFF cells to further confirm the induction of *HSPA6* in the cells (Figure 1e). Then, we detected that the expression of p*HSPA6* in PK cells was significantly higher than that in PK-15 cells (Figure 1b). Next, we used PK cells for passage experiments and observed a significant reduction in F3 cells compared to the expression of p*HSPA6* in F0 cells (Figure 1c). Finally, a cell density assay was performed using PK cells at F0 and three cell culture densities (30%, 80%, and 130%), which showed the highest expression of p*HSPA6* in low-density cell culture (30%) and significantly reduced expression of p*HSPA6* when the cell density reached 80%, which may have been due to inhibition of p*HSPA6* expression when the cell density exceeded 68%. However, when the cell density reached 130%, that is, the cells showed lamination, the expression of p*HSPA6* did not change significantly (Figure 1d).

We concluded that cell type, cell line, generation, and cell density could impact the expression of p*HSPA6*. The results indicated that p*HSPA6* was characterized by unnoticeable constitutive expression but was greatly induced following stress. We thus sought to identify key regulatory elements affecting p*HSPA6* expression.

### 3.2. Effect of HSE-1 on the Expression of pHSPA6

Previous analysis for heat shock element (HSE) detection revealed a site at −100 to −60 bp within the human *HSPA6* promoter [22,30,32]. To validate whether the site has an impact on p*HSPA6* expression, we blasted its sequence against the p*HSPA6* promoter and named it HSE-1; it is located at −73~−107 bp (Figure 2a, Appendix A). Then, we designed a guide RNA (sgRNA-H1) to target HSE-1. Vectors combining sgRNA-H1 with Cas9 were transfected into PFFs using electroporation to assess the efficiency of sgRNA-H1. After eighty-four hours, genomic DNA was isolated from these cells, and Sanger sequencing was performed using PCR products containing the target site. Chromatograms showed obvious multipeaks around the cleavage site. The results suggested that sgRNA-H1 could be used to effectively target HSE-1 in subsequent experiments (Figure 2b).

Previous studies used the *HSPA6* promoter to construct a cell line (H_2_E) that can evaluate cell sensors in response to a stimulus based on EGFP expression [21]. Next, sgRNA-H1-specific PX330 plasmids were injected into H_2_E cells through electroporation. After forty-eight hours, we used a limiting dilution method for injecting the transfected H_2_E cells into 100 mm dishes (1500 cells per plate), and then clones were selected. The TA cloning and Sanger sequencing results were analyzed for indels, which suggested that the genotype of the HSE-1 clone was homozygous −89 T ins (an insertion of ‘T’ at −89 bp) (Figure 2c; Appendix A).

Therefore, we speculated that the p*HSPA6*-89 T ins variant, which destroys an HSF binding site, determines the molecular phenotype. Furthermore, we also examined an association between the variant genotype and the fold change of p*HSPA6* mRNA expression under heat stress, and the results suggested that the mutation could significantly reduce the mRNA expression level of p*HSPA6* (by 75.23%) (Figure 2d). In addition, to explore whether the p*HSPA6*-89 T ins variant would influence the response of p*HSPA6* to other stressors, we treated wild-type and HSE-1 clone cells with copper sulfate, MG132, and curcumin. Our results indicated that the mutation resulted in 99.54%, 99.11%, and 96.95% decreases, respectively (Appendix A). EGFP fluorescence was further confirmed via FACS analysis (Appendix A). 

### 3.3. Effect of HSE-2 on the Expression of pHSPA6

A previous study predicted an HSE at –284 to –265 bp within the human *HSPA6* promoter but did not explain its function [32]. To identify whether the element has an impact on the expression of p*HSPA6*, we blasted its sequence against the p*HSPA6* promoter and named it HSE-2, which is located at –254~–235 bp (Figure 3a, Appendix A). Then, we designed a guide RNA (sgRNA-H2) to target HSE-2 and used the same method we used for HSE-1 to evaluate the targeting efficiency of sgRNA-H2. Sanger sequencing showed obvious multipeaks near the cleavage site of Cas9. Hence, sgRNA-H2 could be used to effectively target HSE-2 in subsequent experiments (Figure 3b).

Furthermore, sgRNA-H2-specific PX330 plasmids were injected into H_2_E cells through electroporation, and clones were selected. The TA cloning and Sanger sequencing results were analyzed for indels, which suggested that the genotype of the HSE-2 clone was homozygous –230 10n del (10 nt deletion at –230 bp) (Figure 3c; Appendix A).

We thus speculated that the p*HSPA6* –230 10n del variant, which destroys an HSF binding site, determines the molecular phenotype. Then, we also examined the association between the variant genotype and the fold change in p*HSPA6* mRNA expression under stressors including heat, copper sulfate, MG132, and curcumin, and the results suggested that the mutation could significantly reduce the mRNA expression level of p*HSPA6* (by 45.00%, 87.19%, 36.27%, and 72.37%, respectively) (Figure 3d, Appendix A). EGFP fluorescence was further confirmed via FACS analysis (Appendix A).

### 3.4. Effect of the CR on the Expression of pHSPA6

Based on homology analysis of seven species, we first found a highly conserved region (CR) located at −646 bp ~ −575 bp within the p*HSPA6* promoter (Figure 4a). To detect whether CR contributes to p*HSPA6* expression, we designed a pair of sgRNAs to knock out the region. Then, we used the same method we used for HSE-1 to evaluate the targeting efficiency of the pair of sgRNAs, and Sanger sequencing revealed obvious deletions between the Cas9 cleavage sites of sgRNA1-1 and sgRNA1-2. Hence, the pair of sgRNAs could be used to effectively target the CR in subsequent experiments (Figure 4b).

Next, the pair of sgRNA-specific PX330 plasmids was injected into H_2_E cells through electroporation, and clones were selected. According to the results of TA cloning and Sanger sequencing, we obtained two genotypes: homozygous −525 183n del (183 nt deletion at –525 bp) (#1) and heterozygous –525 183n del (#2) (Figure 4c; Appendix A). In addition, we examined the insertion of EGFP at a specific site by PCR (Appendix A).

Then, we also detected the fold change in p*HSPA6* mRNA expression under stressors including heat, copper sulfate, MG132, and curcumin, and the results suggested that the #1 clone could significantly reduce the mRNA expression level of p*HSPA6* (by 99.21%, 94.85%, 98.68%, and 94.51%, respectively) and that the #2 clone could significantly reduce the mRNA expression level of p*HSPA6* (by 92.40%, 94.18%, 91.08%, and 88.66%, respectively) (Figure 4d; Appendix A). EGFP fluorescence was further confirmed via FACS analysis (Appendix A).

### 3.5. Effects of the Three Regulatory Regions on the Expression of Other Genes

A previous study confirmed that *HSPA6* induction is likely to link with some members of the HSP40 family [17]. To further determine whether decreased p*HSPA6* expression could alter the expression of HSP40 family members, we measured the expression of some genes related to *HSPA6* in WT and clone cells. The results indicated that *GS* and *DNAJB1* had a positive correlation with downregulated p*HSPA6* in the three clones, and *DNAJC3* had a positive correlation with downregulated p*HSPA6* in clones HSE-1 and HSE-2, whereas *DNAJC3* was uncorrelated with CR (Figure 5a,d–f).

Since HSP70 and HSP90 play an important role in cancer and neurodegenerative diseases, we wondered whether CRISPR/Cas9 editing of HSE-1, HSE-2, and CR affected the expression of p*HSP70.2* and p*HSP90AA1*. Therefore, the expression levels of p*HSPA6*, p*HSP70.2,* and p*HSP90AA1* in the four cell lines were detected. qPCR results determined that the expression of p*HSP70.2* was negatively correlated with changes in p*HSPA6* in clones HSE-1 and HSE-2 but positively correlated with CR, and the expression of p*HSP90AA1* was negatively correlated with changes in p*HSPA6* in clones HSE-1, HSE-2, and CR (Figure 5a–c). Therefore, the expression of p*HSPA6* can affect the expression of p*HSP70.2* and p*HSP90AA1*.

## 4. Discussion

*HSPA6* plays an important role in research on neurodegenerative diseases, tumors, biosensors, and other topics. Pigs have important research value in agriculture, biomedicine, and so on. Currently, most promoter studies of *HSPA6* focus on humans, whereas pigs are studied less frequently. Here, we identified and verified the regulatory sequence of the p*HSPA6* promoter based on H_2_E, which can evaluate cell sensors in response to a stimulus based on EGFP expression. *HSPA6* is basically not expressed in different human cells without stress, but the expression level increases rapidly after stress [34]. Consistent with the results of this study, p*HSPA6* also increased significantly in different porcine cells after stress, while the increases were not consistent among cell types. This may be due to differences in the sensitivity of different types of porcine cells to the same stressor (Figure 1a). Then, we found that compared with that in porcine kidney cell lines, the expression of p*HSPA6* was higher in porcine primary kidney cells after heating. The results of the passage test proved that the subgenerations were related to the expression of p*HSPA6*. Previous research has shown that *HSPA6* expression depends on the number of cells; therefore, we also performed a cell density test. The highest expression of p*HSPA6* was detected in low-density cell culture (30%), and the expression of p*HSPA6* was significantly reduced when the cell density reached 80%, which may have been due to p*HSPA6* expression inhibition when the cell density exceeded 68%. However, when the cell density reached 130%, that is, the cells showed lamination, the expression of p*HSPA6* did not change significantly. Hence, the results were consistent with those of a previous study [35].

The expression of *HSPA6* is strictly induced, which is necessarily related to the regulatory elements in the promoter of *HSPA6*. Previous studies have reported that compared with other HSP70 promoters with basic expression, the upstream promoter elements, TATA box and CAAT box, of *HSPA6* show nucleotide variation, and mutation analysis of these two elements indicated that they are necessary for basic expression of the human HSP70 gene [36]. In addition, it has been determined that RNA polymerase II is suspended on the HSP70 promoter under normal conditions but ceases ahead of activation with stress-induced HSF-1, leading to initiation of transcription and rapid induction [37,38]. When cells or bodies are subjected to physiological stress (such as heat), HSF1 oligomerizes to convert monomers to homotrimers, which translocate to the nucleus and bind to the HSE, thus inducing transcription of heat shock genes [39]. Two HSEs were edited by CRISPR/Cas9 technology, and our findings suggested that nucleotide variation in the two HSEs significantly changed the expression of p*HSPA6*. This may be because the mutation destroys the HSE sequence and prevents HSF from binding to it; thus, it would reduce the promoter activity of *HSPA6* and the expression of the gene. Interestingly, compared with the deletion of 10 bases in the HSE sequence, the insertion of a base (T) at a specific location could reduce p*HSPA6* expression more. Furthermore, CR inhibited most of the promoter activity of p*HSPA6*, suggesting that this region holds a crucial position in the induction of p*HSPA6* expression.

The promoter of *HSPA6* can be used to construct biosensors based on the characteristics of strict induction. The sensitivity of the biosensor can be improved by modifying the regulatory element of the *HSPA6* promoter. A previous study placed CRE and mutated AP-1 in front of the *HSPA6* promoter to improve sensor sensitivity for cadmium chloride [31]. HSE-1, HSE-2, and CR significantly reduced the expression of p*HSPA6*. We might well use these regions to improve the sensitivity of biosensors by replicating them in tandem to overexpress p*HSPA6*.

Various cellular stresses can greatly induce the expression of *HSPA6*, such as mechanical [3], copper and zinc [25], nitric oxide [26,27], heavy metal [28], and antibiotic [29] stress. MG132, as a proteasome inhibitor, can cause protein misfolding, inducing *HSPA6* expression to promote protein refolding [40]. Curcumin, a root extract, has antitumor effects and can also induce *HSPA6* expression. Here, cell lines were treated by heating, copper sulfate, MG132, and curcumin. The results showed that all four treatments could induce p*HSPA6* expression in WT cells, which was considered the same as our previous report [21]. However, after the three regulatory sequences were edited, the expression of p*HSPA6* decreased significantly, indicating that the three regulatory sequences responded not only to heat stress but also to copper sulfate, MG132, and curcumin.

HSP40, HSP70, and HSP90 have been well studied. Hsp40/DnaJ, as a co-partner, helps to achieve the complex functions of HSP70 members. The editing of three regulatory sequences in the p*HSPA6* promoter can reduce the expression levels of *GS*, *DNAJB1,* and *DNAJC3*, but the expression of *DNAJC3* was uncorrelated with the changes in p*HSPA6* in clone CR. These data showed that the downregulation of p*HSPA6* could affect the expression of the genes. This may be because induced HSPA6 interacts with some members of the HSP40 family [17]. HSP70 inhibitors would be a new strategy for enhancing cancer treatment due to the antiapoptotic activity of the HSP70 gene [41]. It is worth noting that these inhibitors have been hampered when used in patients due to toxicity [42]. DNAJA1 acts as a co-partner to regulate the expression of HSP70, and the study suggests that inhibition of DNAJA1 may be a novel anticancer strategy [43]. In addition, targeting the HSP40/HSP70 partner axis may also be feasible for the treatment of castration-resistant prostate cancer [44]. Interestingly, HSP90 is also associated with the development of a variety of cancers, but current HSP90 inhibitors do not have ideal clinical effects [45]. Therefore, HSP40s, *HSP70,* and HSP90 can be considered potential therapeutic targets for cancers. HSP70 and HSP90 also play an important role in neurodegenerative diseases [46,47]. Previous studies have shown that inhibition of HSP90 can promote the expression of *HSAP6* [48], but the effect of *HSPA6* expression on HSP70 and HSP90 has not been studied, and the results of our study help fill this gap. Interestingly, although the editing of all three regulatory sequences reduces the expression of p*HSPA6*, different regulatory regions have different effects on the expression of p*HSP70.2* and p*HSP90AA1*. The expression of p*HSP70.2* was negatively correlated with downregulated p*HSPA6* in clones HSE-1 and HSE-2 but positively correlated with CR, and the expression of p*HSP90AA1* was negatively correlated with downregulated p*HSPA6* in clones HSE-1, HSE-2, and CR (Figure 5a–c). Therefore, three regulatory regions in the p*HSPA6* promoter are potential drug targets for the study of neurodegenerative diseases and cancer. Meanwhile, HSP70 and HSP90 family members have a significant role in animal thermotolerance [49,50]. Thus, these regions contribute to breeding heat-resistant herds.

## 5. Conclusions

In summary, our study reveals two HSEs and a newly discovered CR in the p*HSPA6* promoter by a CRISPR/Cas9-mediated precise gene editing strategy. Gene expression data showed that sequence disruption of these regions could significantly reduce the expression of p*HSPA6* under heat stress. Stimulation studies indicated that these regions responded not only to heat stress but also to copper sulfate, MG132, and curcumin. Among these regions, the knockout of CR could more strongly reduce the expression of p*HSPA6* under stressors. Further mechanistic studies showed that downregulated p*HSPA6* could significantly affect some important members of the HSP family that are involved in heat resistance and many diseases. Therefore, our novel results may provide a potential therapeutic strategy for neurodegenerative diseases and cancers, as well as beneficial strategies for breeding heat-resistant herds and research on biosensors.

## Figures and Tables

**Figure 1 genes-13-00189-f001:**
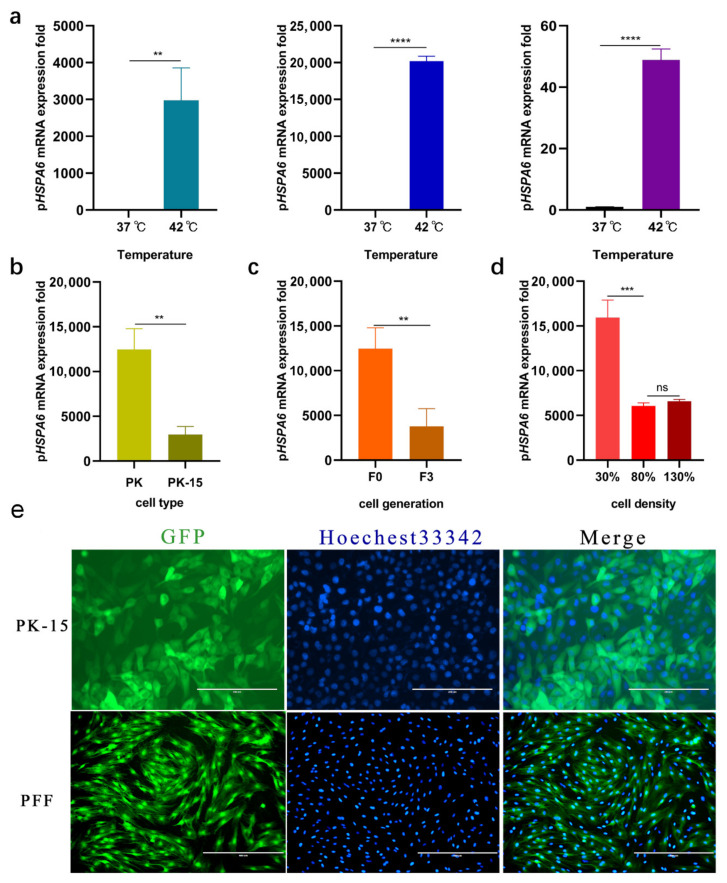
Analysis of the *HSPA6* gene in porcine cells. (**a**) The mRNA expression of *HSPA6* in PK-15, PFF, and 3D4/2 cells was detected by qPCR under heat. ** *p <* 0.01, **** *p <* 0.0001. (**b**) qPCR analysis of inducible p*HSPA6* expression in PK and PK-15 cells. ** *p <* 0.01. (**c**) qPCR analysis of inducible p*HSPA6* expression in F0 and F3 cells. ** *p <* 0.01. (**d**) qPCR analysis of the cell density of inducible p*HSPA6* expression in PK cells. *** *p <* 0.001, ns = not significant. (**e**) Fluorescence microscopy analysis of the eGFP expression in PK-15 cell line and PFF cells under heat.

**Figure 2 genes-13-00189-f002:**
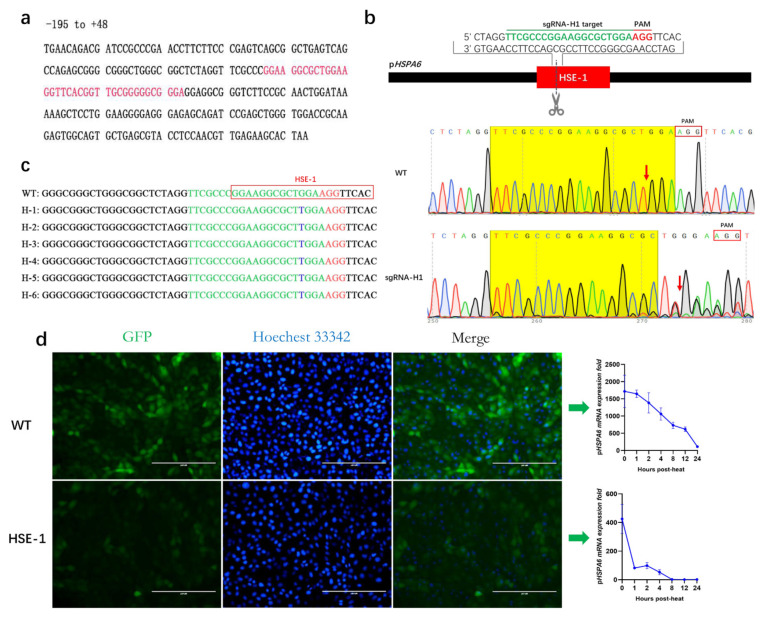
Effect of HSE-1 on the expression of p*HSPA6*. (**a**) The position of HSE-1 was located in the p*HSPA6* promoter. (**b**) Strategy of sgRNA-H1 knock-in into p*HSPA6*; the cutting efficiency of sgRNA-H1 was evaluated using chromatograms. Red blocks and red arrows represent PAMs and cleavage sites, respectively. (**c**) Diagram of positive clone HSE-1 after pLB vector cloning. (**d**) Fluorescence microscopy and qPCR analysis of p*HSPA6* expression in WT and positive clone cells (data show the mean ± S.E.M. in triplicate assay).

**Figure 3 genes-13-00189-f003:**
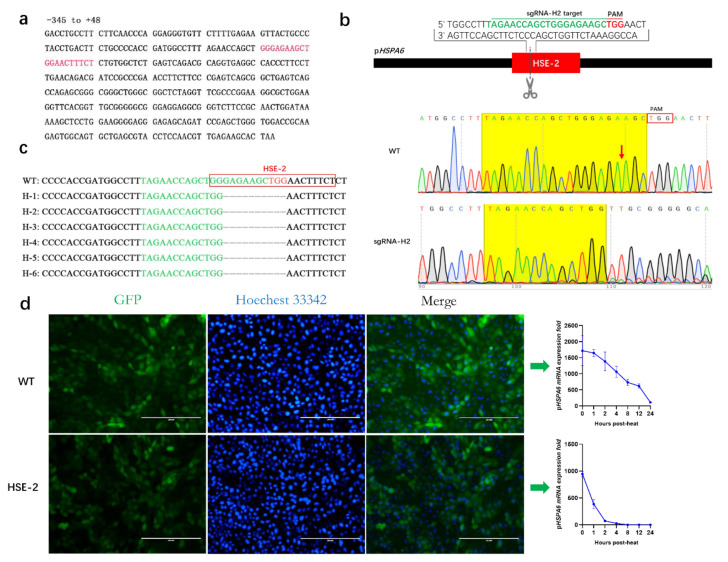
Effect of HSE-2 on the expression of p*HSPA6*. (**a**) The position of HSE-2 was located in the p*HSPA6* promoter. (**b**) Strategy of sgRNA-H2 knock-in into p*HSPA6*; and the cutting efficiency of sgRNA-H2 was evaluated using chromatograms. Red blocks and red arrows represent PAMs and cleavage sites, respectively. (**c**) Diagram of positive clone HSE-2 after pLB vector cloning. (**d**) Fluorescence microscopy and qPCR analysis of p*HSPA6* expression in WT and positive clone cells (data show the mean ± S.E.M. in triplicate assay).

**Figure 4 genes-13-00189-f004:**
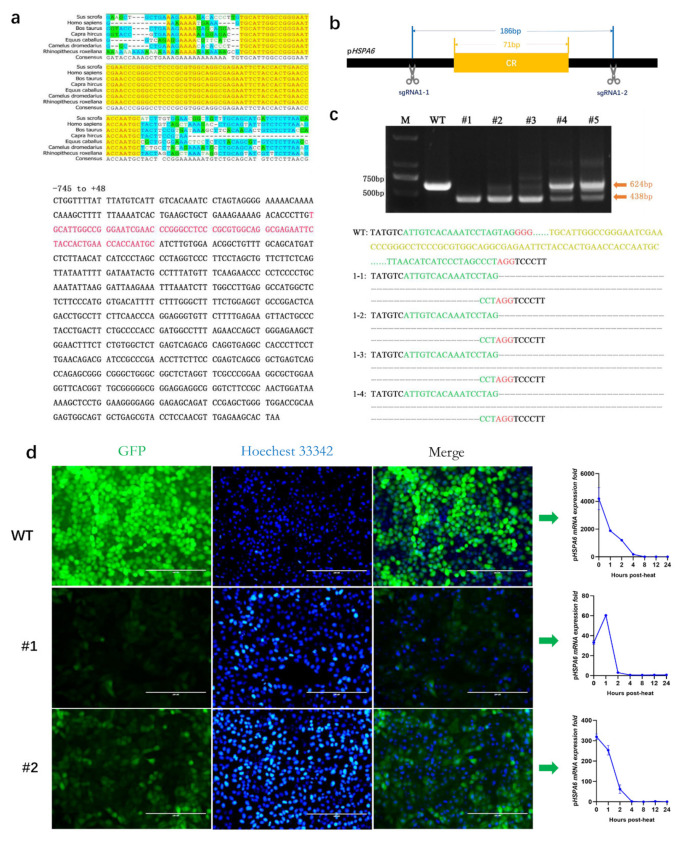
Effect of CR on the expression of p*HSPA6*. (**a**) CR was found by blasting the *HSPA6* promoter of seven species, and the position of CR was located in the p*HSPA6* promoter. (**b**) Strategy of sgRNA1-1 and sgRNA1-2 knock-in into p*HSPA6*. (**c**) Selecting of positive clones using PCR. Primers are shown in Table 2; diagram of positive clone #1 after pLB vector cloning. (**d**) Fluorescence microscopy and qPCR analysis of p*HSPA6* expression in WT and positive clone cells (data show the mean ± S.E.M. in triplicate assay).

**Figure 5 genes-13-00189-f005:**
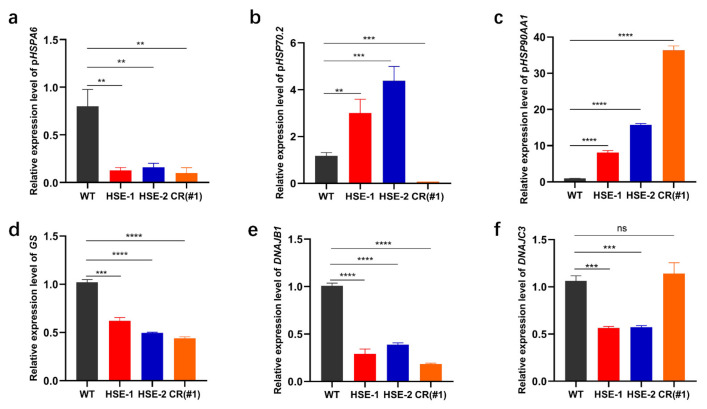
Effects of the three regulatory regions on the expression of other genes after thermal treatment. (**a**) HSE-1, HSE-2, and CR significantly reduced the expression of p*HSPA6*. ** *p <* 0.01. (**b**) The expression of p*HSP70.2* was negatively correlated with changes in p*HSPA6* in clones HSE-1 and HSE-2 but positively correlated with CR. ** *p <* 0.01, *** *p <* 0.001. (**c**) The expression of p*HSP90AA1* was negatively correlated with changes in p*HSPA6* in clones HSE-1, HSE-2, and CR. **** *p <* 0.0001. (**d**) The expression of *GS* was positively correlated with changes in p*HSPA6* in clones HSE-1, HSE-2, and CR. *** *p <* 0.001, **** *p <* 0.0001. (**e**) The expression of *DNAJB1* was positively correlated with changes in p*HSPA6* in clones HSE-1, HSE-2, and CR. **** *p <* 0.0001. (**f**) The expression of *DNAJC3* was positively correlated with changes in p*HSPA6* in clones HSE-1 and HSE-2 but uncorrelated with CR. *** *p <* 0.001, ns = not significant.

**Table 1 genes-13-00189-t001:** Sequences of sgRNAs.

Name	Primer	Sequence (5′–3′)
sgRNA-H1	sgRNA-H1-F	CACCGTTCGCCCGGAAGGCGCTGGA
	sgRNA-H1-R	AAACTCCAGCGCCTTCCGGGCGAAC
sgRNA-H2	sgRNA-H2-F	CACCGTAGAACCAGCTGGGAGAAGC
	sgRNA-H2-R	AAACGCTTCTCCCAGCTGGTTCTAC
sgRNA1-1	sgRNA1-1-F	CACCGATTGTCACAAATCCTAGTAG
	sgRNA1-1-R	AAACCTACTAGGATTTGTGACAATC
sgRNA1-2	sgRNA1-2-F	CACCGTTAACATCATCCCTAGCCCT
	sgRNA1-2-R	AAACAGGGCTAGGGATGATGTTAAC

**Table 2 genes-13-00189-t002:** Primers used in this study.

Name	Primer	Sequence (5′–3′)	Usage
HSE-1-KO	H1-F	CTCTCTTCCCATGGTGA	PCR
	H1-R	GCTGGTGCATCTGACTTCAT	
HSE-2-KO	H2-F	CTCTCTTCCCATGGTGA	PCR
	H2-R	GCTGGTGCATCTGACTTCAT	
CR-KO	F1-F	CCTTTCTGGGCTGCGACTTGAT	PCR
	R1-R	GGGCGGATCGTCTGTTCAAGGA	
EGFP-KI [21]	EGFP-KI -F	GAGGCGCATGTTCTCCAAAAACC	PCR
	EGFP-KI -R	AGCCACACTTGTAGTTGCACTGG	
p*HSPA6* [21]	p*HSPA6* q-F	ATCCATGATATTGTCCTA	qPCR
	p*HSPA6* q-R	TTATGCTCTTGTTCAGTT	
GS [21]	GS q-F	CTTGCATCGTGTGTGCGAAG	qPCR
	GS q-R	GCTTAGCTTCTCGATGGCCT	
DNAJB1 [21]	B1 q-F	TGACCATCGAAGTGAAGCGG	qPCR
	B1 q-R	TCGGCTGGAATGTTGTTGGA	
DNAJC3 [21]	C3 q-F	GGAGCCTGACAATGTGAATGC	qPCR
	C3 q-R	GACCTTCTCGAATCTGCTGGT	
p*HSP70.2*	70 q-F	GAGCAAGGAGGAGATCGAGC	qPCR
	70 q-R	GTTGAAGGCGTACGACTCCA	
p*HSP90AA1*	90 q-F	TCGAAGGGCAGTTGGAGTTC	qPCR
	90 q-R	ATGAGCTCCTCGCAGTTGTC	

## Data Availability

Data are contained within the article or Appendix A.

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
