# Peer review of "Identification and Functional Analysis of the Regulatory Elements in the pHSPA6 Promoter"

_genes, 2022, doi:10.3390/genes13020189_

Round 1

Reviewer 1 Report

In this article, author studied the stress sensitive sites in porcine HSPA6 (pHSPA6) and identified two heat shock elements (HSEs) and a conserved region (CR) in the pHSPA6 promoter. They demonstrated that sequence disruption of these regions could significantly reduce the expression of pHSPA6 under heat stress. Further mechanism studies showed that downregulated pHSPA6 could significantly affect some important members of HSP family which are involved in Hsp40, Hsp70, and HSP90.

Overall, the study is well written using the advanced technologies, however additional experimental approach are needed to support hypothesis.

Other minor comments:

1: from where cell lines were obtained?

2: Higher quality images should be used.

3: There are several typo errors that should be corrected, example

Page 1, line 42: “for example, T cells were engineered can control gene expression remotely “

Author Response

Response to Reviewer 1 Comments
Dear Reviewer:
Thank you very much for your comments concerning our manuscript entitled
“Identification and functional analysis of the regulatory elements in the pHSPA6
promoter” (genes-1521058). Those comments are all valuable and very helpful for
revising and improving our manuscript, as well as the important guiding significance to our researches. We have made extensive modifications on the original manuscript in accordance with the reviewer’ comments. In our revised manuscript, these revised portions are marked in yellow and extensive modifications were marked using “Track Changes”. Here below are our point-by-point responses to reviewer’ s comments.
Point1: In this article, author studied the stress sensitive sites in porcine HSPA6
(pHSPA6) and identified two heat shock elements (HSEs) and a conserved region (CR) in the pHSPA6 promoter. They demonstrated that sequence disruption of these regions could significantly reduce the expression of pHSPA6 under heat stress. Further mechanism studies showed that downregulated pHSPA6 could significantly affect some important members of HSP family which are involved in Hsp40, Hsp70, and HSP90.
Overall, the study is well written using the advanced technologies, however additional experimental approach are needed to support hypothesis.
Response1: Thanks for your helpful and kindly advise. As your suggestion, the new data of fluorescence analysis and flow cytometry have been introduced in
page 4, line 4; page 7, line 2; page 8, line 16 and page 10, line 12. Please see our
revised manuscript for details.

Point2: from where cell lines were obtained?

Response2: We bought Porcine kidney cell line-15 (PK-15) cells (Lot Number:
58808810 ATCC Number: CCL-33). Previously, pHSPA6-based EGFP PK-15
fluorescent reporter cell line (H2E) was designed and developed via a CRISPR/Cas9-mediated knock-in strategy. Relative descriptions have been introduced in our manuscript. In addition, Porcine primary kidney cells (PK) were isolated from 2-dayold fetuses, Porcine fetal fibroblasts (PFFs) were isolated from 33-day-old fetuses and porcine alveolar macrophage cell lines (3D4/2) were obtained from Prof. Li Feng of Harbin Veterinary Research Institute. We have highlighted the information. Please see page 17, line 3 for details.

Point3: Higher quality images should be used.

Response3: Thank you very much for your kindly advices. We have combined the
original images, adjusted the resolution of the images to more than 600dpi, and changed the color scheme, font size and other information of the figures. Please see the manuscript for details.

Point4: There are several typo errors that should be corrected, example
Page 1, line 42: “for example, T cells were engineered can control gene expression remotely”
Response4: Thank you very much for your kindly comments and we are very sorry for our negligence of writing. This spelling mistake “for example, T cells were engineered can control gene expression remotely” has been corrected to “For example, engineered T cells can control gene expression remotely” in Page 2, line 25 of our revised manuscript. In addition, We have made extensive modifications on the original manuscript in accordance with your comments, carefully proof-read the manuscript to minimize typographical, grammatical and bibliographical errors and the English writing in our revised manuscript has been improved by the highly qualified native English speaking editors at AJE (the verification code: 7E9C-DE71-A42D-CD6BC73P).

Reviewer 2 Report

This manuscript bu Jiao et. al., identified two heat shock elements and a conserved region in porcine HSPA6 using CRISPR/Cas9. They further test the expression levels of other chaperones in their mutant clones for these regions.

Here are my concerns:

  1. The manuscript has a lot of grammatical errors. Some of the sentences are not clear to the readers.
  2. In the introduction, the authors need to discuss the impact of post translational modifications of HSPA6 on its function. Please refer to  PMID: 34025620 and PMID: 32518165.
  3. Can the authors test the functionality of their mutant clones?
  4. In figure 5 the authors just show the expression levels of other co-chaperones in the mutant clones. These level changes just depict correlation. Can the authors show the level of interaction of these mutants and these chaperones using immunopecipitation?
  5. In the discussion, line 267, the authors mention use of chaperone inhibitors for cancer treatment. The authors should also discuss targeting co-chaperones fore cancer treatment. Please refer PMID:32796891 and PMID:29764864.
  6. Some of the graphs in the figures are not readable due to light colors. Please change the colors to make them readable to the audience.

Author Response

Response to Reviewer 2 Comments
Dear Reviewer:
Thank you very much for your comments concerning our manuscript entitled
“Identification and functional analysis of the regulatory elements in the pHSPA6
promoter” (genes-1521058). Those comments are all valuable and very helpful for
revising and improving our manuscript, as well as the important guiding significance to our researches. We have made extensive modifications on the original manuscript in accordance with the reviewer’ comments. In our revised manuscript, these revised portions are marked in yellow and extensive modifications were marked using “Track Changes”. Here below are our point-by-point responses to reviewer’ s comments.
Point1: The manuscript has a lot of grammatical errors. Some of the sentences are not clear to the readers.
Response1: Thanks a lot for your patient and careful reading of our manuscript. We are sorry for our negligence of writing. We have made extensive modifications on the original manuscript in accordance with your comments, carefully proof-read the manuscript to minimize typographical, grammatical and bibliographical errors and the English writing in our revised manuscript has been improved by the highly qualified native English speaking editors at AJE (the verification code: 7E9C-DE71-A42DCD6B- C73P).
Point2: In the introduction, the authors need to discuss the impact of post translational modifications of HSPA6 on its function. Please refer to PMID: 34025620 and PMID: 32518165.
Response2: We appreciate so much the very helpful comments. As your suggestion, We have discussed the impact of post translational modifications of HSP70s on its function according to PMID: 32518165, please see the highlighted section (page 2, line 3) in the introduction for details. Meanwhile, we have also discussed the role of HSPA6 during Enterovirus A71 Infection according to PMID: 34025620, please see the highlighted section (page 2, line 19) in the introduction for details.
Point3: Can the authors test the functionality of their mutant clones?
Response3: Thanks for your kindly advise.
After a careful consideration and discussion, we think it’s unnecessary to test the
functionality of their mutant clones, as the present study focused on identifying the regulatory elements in the pHSPA6 promoter and verifying its effect on gene
expression. In the future, different stress stimulation and multispecies derived cells should be considered to further explore the functionality of their mutant clones.
Point4: In figure 5 the authors just show the expression levels of other co-chaperones
in the mutant clones. These level changes just depict correlation. Can the authors show the level of interaction of these mutants and these chaperones using
immunoprecipitation?
Response4: Thanks for your precious advice and it will definitely help to improve the integrity of our manuscript. During the initial research period, western blot or
immunoprecipitation assays was used to examine the effects of downregulated pHSPA6 on some important members of HSP family in the protein level. However, currently limited to the effectiveness and specificity of commoditized porcine antibody, that is difficult to provide more empirical evidences from protein levels. In order to further improve the integrity of our manuscript, the new data of fluorescence analysis and flow cytometry have been introduced in page 4, line 4; page 7, line 2; page 8, line 16 and page 10, line 12. Please see our revised manuscript for details. In the future, to carry on the system more in-depth study, expression of specific genes and development of efficient antibodies against these related porcine proteins is important.
Point5: In the discussion, line 267, the authors mention use of chaperone inhibitors for cancer treatment. The authors should also discuss targeting co-chaperones fore cancer treatment. Please refer PMID:32796891 and PMID:29764864.
Response5: We appreciate so much the very helpful comments. As your suggestion, We have discussed targeting co-chaperones fore cancer treatment according to PMID:32796891 and PMID:29764864, please see the highlighted section (page 15, line 23) in the discussion for details.
Point6: Some of the graphs in the figures are not readable due to light colors. Please change the colors to make them readable to the audience.
Response6: Thank you very much for your kindly advices. We have combined the
original images, adjusted the resolution of the images to more than 600dpi, and changed the color scheme, font size and other information of the figures. Please see the manuscript for details.

Round 2

Reviewer 1 Report

accept

Reviewer 2 Report

The revised manuscript by Jiao et al., has been substantially improved. The authors have provided satisfactory justifications for all my concerns. Therefore, I recommend the manuscript be accepted in the present form.